# Copper Nitride: A Versatile Semiconductor with Great Potential for Next-Generation Photovoltaics

M. I. Rodríguez-Tapiador [1,*] , J. M. Asensi [2,3] , M. Roldán [4] , J. Merino [5], J. Bertomeu [2,3] and S. Fernández [1,*]

1   Energy Department, Center for Energy, Environmental and Technological Research (CIEMAT), Av. Complutense 40, 28040 Madrid, Spain
2   Departament de Física Aplicada, Universitat de Barcelona, 08028 Barcelona, Spain; jmasensi@ub.edu (J.M.A.); jbertomeu@ub.edu (J.B.)
3   Institute of Nanoscience and Nanotechnology (IN2UB), Universitat de Barcelona, 08028 Barcelona, Spain
4   National Fusion Laboratory, CIEMAT, Av. Complutense 40, 28040 Madrid, Spain; marcelo.roldan@ciemat.es
5   Technology Support Center (CAT), University Rey Juan Carlos, Tulipán, s/n, Móstoles, 28039 Madrid, Spain; jesus.merino@urjc.es
*   Correspondence: mariaisabel.rodriguez@ciemat.es (M.I.R.-T.); susanamaria.fernandez@ciemat.es (S.F.); Tel.: +34-913466591 (M.I.R.-T.); +34-913460923 (S.F.)

**Abstract:** Copper nitride ($Cu_3N$) has gained significant attention recently due to its potential in several scientific and technological applications. This study focuses on using $Cu_3N$ as a solar absorber in photovoltaic technology. $Cu_3N$ thin films were deposited on glass substrates and silicon wafers via radio-frequency magnetron sputtering at different nitrogen flow ratios with total pressures ranging from 1.0 to 5.0 Pa. The thin films' structural, morphology, and chemical properties were determined using XRD, Raman, AFM, and SEM/EDS techniques. The results revealed that the $Cu_3N$ films exhibited a polycrystalline structure, with the preferred orientation varying from 100 to 111 depending on the working pressure employed. Raman spectroscopy confirmed the presence of Cu-N bonds in characteristic peaks observed in the 618–627 $cm^{-1}$ range, while SEM and AFM images confirmed the presence of uniform and smooth surface morphologies. The optical properties of the films were investigated using UV-VIS-NIR spectroscopy and photothermal deflection spectroscopy (PDS). The obtained band gap, refractive index, and Urbach energy values demonstrated promising optical properties for $Cu_3N$ films, indicating their potential as solar absorbers in photovoltaic technology. This study highlights the favourable properties of $Cu_3N$ films deposited using the RF sputtering method, paving the way for their implementation in thin-film photovoltaic technologies. These findings contribute to the progress and optimisation of $Cu_3N$-based materials for efficient solar energy conversion.

**Keywords:** $Cu_3N$ films; reactive magnetron sputtering; photothermal deflection spectroscopy (PDS); solar energy conversion





## 1. Introduction

Copper nitride ($Cu_3N$) is a compound that has attracted increasing attention in recent years due to its unique properties and potential applications in several fields [1], as shown in Figure 1.

Regarding its crystal structure, $Cu_3N$ is a binary compound made up of copper (Cu) and nitrogen (N) that displays a cubic anti-$ReO_3$ crystal system (space group Pm3m). The bond angle of Cu-N-Cu reported is approximately 180 degrees, and the lattice parameter and density are 3.817 Å and 6.1 $g/cm^3$, respectively [2]. This material is metastable and exhibits thermal stability close to 250 °C [3]. $Cu_3N$ displays insulator-to-conductor transition behaviour, with electrical conductivity values ranging between $10^{-3}$ and $10^{-2}$ $Scm^{-1}$ [4]. Moreover, the electrical properties of the $Cu_3N$ films can be modified by doping them with metal or non-metal elements such as fluorine or gold, resulting in either p-type or n-type behaviour [5,6]. To switch between these electric characters, doping concentration, synthesis

conditions, surface modifications, and interface engineering can be adjusted to alter charge carriers, crystal structure, defect density, and electrical properties. By controlling these factors, copper nitride's resistivity can be fine tuned to achieve the desired conductivity [7–9]. This approach can potentially enhance and broaden the scope of applications for $Cu_3N$ films. As a result, it is a promising candidate for future technological advancements, developments, and innovations in various fields, including electronic devices such as thin-film transistors (TFTs) and complementary metal-oxide-semiconductor (CMOS) circuits [10].

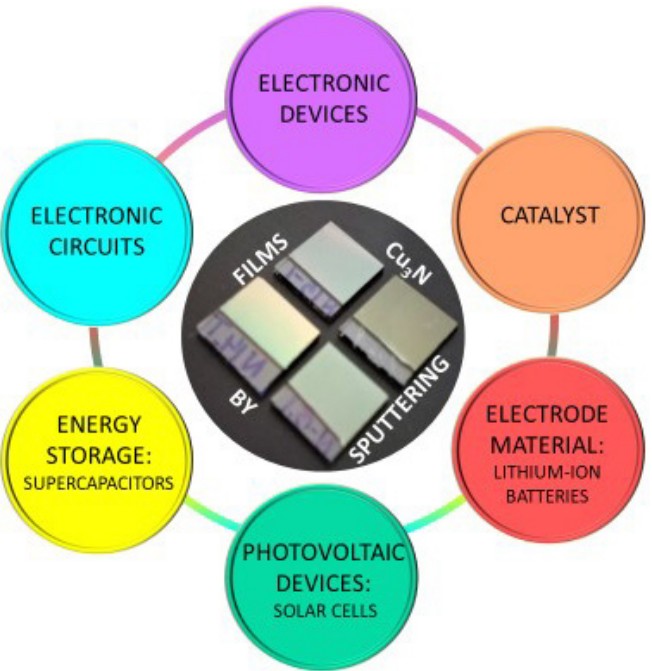

**Figure 1.** Some applications of $Cu_3N$ films in different technological fields.

Additionally, $Cu_3N$ has been investigated as a potential material for energy storage applications, such as batteries and other energy storage devices, due to its thermal stability and chemical stability [11]. Recent works have reported that $Cu_3N$ can be used as an electrode material in lithium-ion batteries, exhibiting excellent performance in capacity and cycle stability [12]. Moreover, copper nitride has demonstrated high electrochemical and catalytic activity for various important reactions, making it a promising material for numerous applications [13–15].

From the optical point of view, experimental studies have shown that this material has indirect and direct band gap values ranging from 1.17 to 1.69 eV and 1.72 to 2.38 eV, respectively [16,17]. Considering these values, $Cu_3N$ can be considered a promising light absorber material for solar cells. Its crystal structure and chemical composition can be optimised by controlling the technical parameters during preparation, resulting in an optimised optical bandgap and maximum photovoltaic voltage [4,18,19]. By developing p-type and n-type $Cu_3N$ (100) thin films via different technologies and adjusting the Cu/N chemical composition via reasonable control methods, bipolar doping can be added under Cu defects, leading to materials ready for photovoltaic applications thanks to their excellent indirect bandgap values. This approach can significantly increase the conversion efficiency of solar energy [4,20].

There are several chemical and physical methods to prepare copper nitride films. Among the chemical fabrication techniques, chemical vapour deposition (CVD) and atomic layer deposition (ALD) are commonly used to fabricate it in a gas phase. These methods involve the use of precursors such as $Cu(hfac)_2$ and $[Cu(sBu-Me-amd)]_2$ [21] to determine its resulting phase composition and morphology and to establish the growth rate [22,23]. Ammonolysis reactions can also be used to prepare bulk $Cu_3N$ powder samples [24], as

well as thin films [25]. In addition, recent works have shown that $Cu_3N$ nanowire arrays can be synthesised by an ammonolysis reaction from copper (II) oxide precursors grown on copper surfaces deposited by electro or PVD in an ammonia solution [26,27].

Regarding the physical ones, sputtering is one of the most popular physical vapour deposition (PVD) methods used to fabricate films of metals, alloys, oxides, and nitrides [28–31]. Since the pioneering work of Terada et al. (1989) on epitaxial growth of copper nitride [32], reactive RF magnetron sputtering has become the most widely used mode for the fabrication of binary nitride. This method involves using a vacuum chamber in which a copper target is bombarded with high-energy ions, causing copper atoms to be ejected from the target and deposited onto a substrate to form a thin film. By introducing nitrogen gas, with or without argon gas, into the chamber during this process, $Cu_3N$ can be grown on a substrate. This method's ease, simplicity, low cost, reproducibility and sustainability make it a very attractive choice for the growth of $Cu_3N$ thin films. Previous studies have already reported that by modifying the bias voltage [33], the type of substrate [32], the working pressure [34], and the RF power [35], the film properties can be adjusted, allowing the variation in optical, electrical, structural, and morphological features to suit them to the desired ones depending on the application field. In addition, in our previous works, we have demonstrated the strong effect of RF power on modifying the morphological, structural, and optical $Cu_3N$ characteristics, intending to use them as solar absorbers for next-generation photovoltaics [19].

In this work, we investigated the impact of the process gas and its pressure on the properties of $Cu_3N$ films prepared via reactive RF magnetron sputtering at room temperature (RT). Two different atmospheres were studied: an environment based on the mixture of $N_2$ and Ar gases and another one based on a pure $N_2$ gas, while the working pressures ranged from 1.0 to 5.0 Pa. The changes in crystalline nature, chemical composition, morphology, and electrical and optical properties were examined in depth. We aim to determine what material and sputtering deposition conditions lead to the most suitable properties and the highest possible absorption coefficient to be used as an absorber in a solar cell.

## 2. Materials and Methods

$Cu_3N$ thin films were deposited on different substrates, <100> polished n-type floating zone crystalline silicon (c-Si) wafers and 1737F Corning glass (Corning Inc., New York, NY, USA) via reactive RF magnetron sputtering in a commercial MVSystem LLC (Golden, CO, USA) mono-chamber sputtering system. The 3-inch diameter Cu target, with a purity of 99.99%, was from Lesker company (St. Leonards-on-Sea, East Sussex, UK). Before the sputtering deposition, the surface of the silicon wafer was prepared by removing the native silicon dioxide layer using a solution of 1% hydrofluoric acid (HF) in a mixture of deionised water and isopropyl alcohol. The wafer was immersed in this solution for 5 min. Next, the glass substrates were subjected to ultrasonic cleaning with ethanol and deionised water for 3 min. Then, they were submerged in isopropyl alcohol. Afterwards, all substrates were dried by blowing nitrogen gas over them.

The sputtering chamber was initially pumped to a base pressure of $2.6 \times 10^{-5}$ Pa, and the distance between the target and substrate was set to 10 cm. A pre-sputtering process was performed for 5 min to clean the target surface. Then, the deposition was conducted for 30 min at room temperature (RT) and 50 W of RF power. The process gases used were $N_2$ (99.9999%) and Ar (99.9999%), with flow rates of 20 sccm and 10 sccm, respectively, controlled using mass flow controllers from MKS Instruments (MKS Instruments, Andover, MA, USA). The total gas pressure was varied between 1.0 and 5.0 Pa by adjusting the position of the "butterfly" valve in the magnetron system. The thickness of the films was measured using a Dektak 8 profilometer (Bruker, San José, CA, USA). A tip force of 68.67 µN and a scan size of 2000 µm were used in all cases. To determine the crystallinity of the $Cu_3N$ films, X-ray diffraction (XRD) was performed using a commercial system (model PW3040/00 X'Pert MPD/MRD) (Malvern Panalytical Ltd., Malvern, UK) with Cu-kα radiation (λ = 0.15406 nm). The 2θ range scanned was 10–60°, with a step size

of 0.01° and a time of 20 s per step. The topography was analysed with a multimode nanoscope atomic force microscopy (AFM) model IIIA (SPM; Veeco Digital Instrument) in tapping mode using a silicon nitride AFM tip (OTR8, Veeco, Santa Clara, CA, USA). The surface roughness was quantified using mean root square (RMS) analysis, and the grain size from the two-dimensional (2D) AFM $1 \times 1$ μm$^2$ images, using the Gwyddion software (Gwyddion software, http://gwyddion.net/ accessed on 20 March 2023).

In addition, the variation of surface morphology as a function of the gaseous environment used was determined using a JEOL JSM 7600F scanning microscope, equipped with a field emission Schottky electron gun (FESEM), in-lens secondary electron detector, and elemental analysis system for chemical composition EDS (energy-dispersive X-ray spectroscopy). Several surface regions were analysed at an acceleration voltage of 15 kV to quantitatively determine the amount of Cu and N in the thin film.

The molecular structure was determined using a dispersive spectrometer Confocal Raman microscope with capabilities for obtaining XYZ 3D confocal Raman images, equipped with a 532 nm laser, two diffraction gratings (600 and 1800 g/nm), three objectives ($5\times$, $75\times$ and $100\times$), and option to obtain photocurrent mappings (Horiba LabRam soleil, Longjumeau Cedex, France). This measurement provides valuable information about the sample's vibrational modes and helps characterise its molecular structure. Finally, to determine the suitability of $Cu_3N$ as a solar absorber, the optical transmittance spectra were measured at normal incidence using a UV/VIS/NIR Perkin Elmer Lambda 1050 spectrophotometer. The optical band gap energies ($E_g$) were calculated from these spectra for indirect and direct transitions.

The optical properties of $Cu_3N$ thin films deposited on glass were analysed using UV-VIS-NIR optical spectroscopy and photothermal deflection spectroscopy (PDS). The transmittance ($T_{opt}$) and reflectance ($R_{opt}$) spectra were obtained with a PerkinElmer Lambda 950 UV-Vis-NIR spectrometer equipped with an integrating sphere. While the transverse PDS setup used to measure weaker absorption consists of a 100 W tungsten halogen lamp, PTI 01-0002 monochromator (spectral range of 400–2000 nm), and Thorlabs MC1000 optical chopper (4 Hz light modulation frequency). A Signal Recovery 7265 lock-in amplifier was connected to a Hamamatsu C10442-02 PSD position-sensitive detector to measure the deflection of an MC6320C 10 mW laser probe beam. Samples were put in a quartz cell filled with Fluorinert TM FC-40. $T_{opt}$ and $R_{opt}$ measurements allow for determining the optical absorbance ($A_{opt} = 1 - T_{opt} - R_{opt}$) in the strong absorption region. In addition, the interference fringes observed in both spectra can be used to estimate the film thickness and their refractive index ($n_\infty$).

On the other hand, the PDS measurement is very effective for determining absorbance (APDS) in the weak (and very weak) absorption region. Thus, it is possible to determine the absorbance over a broad spectrum range by combining optical measurements with PDS. The absorption coefficient ($\alpha$) is obtained by a fit based on the calculation of absorbance using the transfer matrix method (TMM). These measurements allow the determination of the most suitable sputtering conditions to achieve a more efficient solar absorber material.

## 3. Results

The $Cu_3N$ films in this study exhibited excellent physical stability and good adhesion to the respective substrate, even after exposure to ambient air; therefore, no evidence of cracking or peeling off was observed after the deposition process. Table 1 details the sputtering deposition conditions and the corresponding measured film thickness.

As observed, the deposition rate varied depending on the deposition conditions used, obtaining similar values to those reported by other authors [32,36]. These data reveal that the working pressure is essential in thin film fabrication. As the working pressure increases, the number of impacts between the sputtered species and gas atoms increases in sputtering processes. Thus, we observed that the deposition rate decreases with higher working pressure. This effect leads to a decrease in the deposition rate due to a reduction in the mean free path of the species within the plasma [3]. This trend was observed for the

samples deposited in the $N_2$ pure atmosphere, as shown in Figure 2. It is well known that there is an abrupt decrease in deposition rate due to the nitridation process that is more favoured at high gas pressures [37]. At the same time, the sputtering ratio Cu/N began to be higher when the $N_2$ gas pressure decreased. This would favour the metallic regime and, hence, obtain a gradual increase in the sputtering yield at low working pressures. In addition, this would indicate that, under such values of $N_2$ gas pressure, the target "poisoning" effect would not have started yet; hence, a gradual decrease in the deposition rate with the gas pressure was achieved.

**Table 1.** Deposition conditions of $Cu_3N$ films, varying $N_2$ flow ratios (0.7 and 1.0) and different total pressures.

| Total Pressure (Pa) | $N_2$ Flux | Ar Flux | $N_2$ Flow Ratio | Deposition Rate (nm/s) | Thickness (nm) |
|---|---|---|---|---|---|
| 1.0 | 20 | 10 | 0.7 | 0.053 | $95 \pm 5$ |
| 2.0 | 20 | 10 | 0.7 | 0.054 | $96 \pm 7$ |
| 3.5 | 20 | 10 | 0.7 | 0.115 | $207 \pm 18$ |
| 5.0 | 20 | 10 | 0.7 | 0.055 | $98 \pm 2$ |
| 1.0 | 20 | 0 | 1.0 | 0.091 | $164 \pm 12$ |
| 2.0 | 20 | 0 | 1.0 | 0.065 | $118 \pm 20$ |
| 3.5 | 20 | 0 | 1.0 | 0.060 | $109 \pm 16$ |
| 5.0 | 20 | 0 | 1.0 | 0.058 | $104 \pm 20$ |

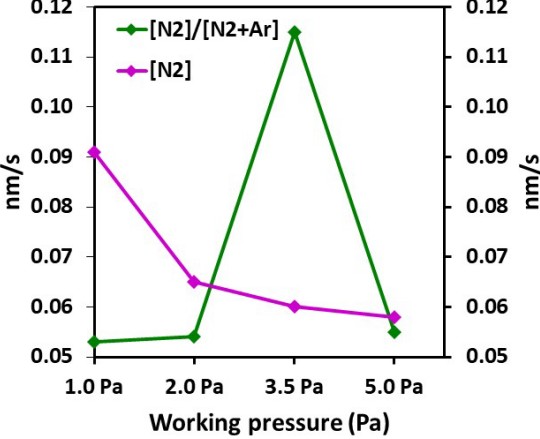

**Figure 2.** Deposition rate to different $N_2$ flow ratios of 0.7 and 1.0 and to different total working pressure (Pa). Deposition conditions of $Cu_3N$ films: RT and $P_{rf} = 50$ W.

On the other hand, it can be noticed that the films prepared under the $N_2$/Ar gas mixture environment presented lower deposition rates than those prepared in $N_2$ pure atmosphere. In the case where the sputtering process was carried out in a $N_2$/Ar atmosphere, the decrease in the deposition rate was not so evident, remaining almost constant at 0.055 nm/s (see Figure 2). This could be attributed to a poorer nitridation occurring on the surface target due to the lower presence of $N_2$ in the gas mixture. It should be pointed out that a significant increase was observed in the deposition rate at 3.5 Pa, reaching a value of 0.115 nm/s. This was attributed to the change in the preferred crystal structure orientation, specifically to the (111) plane, a plane of lower density that would lead to a rougher film with a lower refractive index, as will be shown later [3].

Figure 3 illustrates the X-ray diffraction (XRD) patterns of the $Cu_3N$ deposited at the operating pressure range of 1.0–5.0 Pa in different environments: an $N_2$ pure atmosphere (Figure 3a) and a gas mixture of $N_2$/Ar (Figure 3b). All of the films exhibited a polycrystalline nature, characterised by an anti-$ReO_3$ crystal structure, typical of cubic $Cu_3N$

(card number 00-047-1088), and hence, dominated with the $Cu_3N$ phase. Regardless of the working gas pressure used, the samples deposited in the $N_2$ pure atmosphere showed the (100) plane as the preferred orientation (Figure 3a).

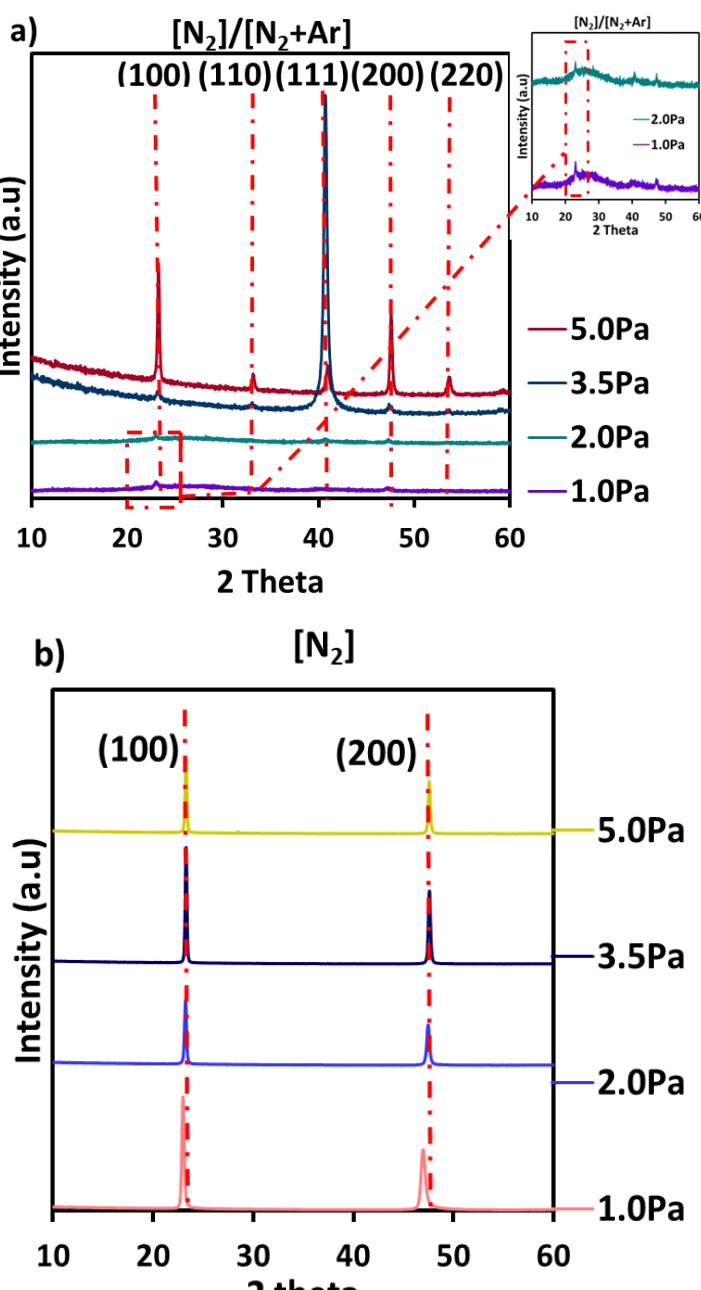

**Figure 3.** X-ray diffraction spectra of $Cu_3N$ films at different gas pressures (Pa) and nitrogen flow ratios. (**a**) $[N_2]/[N_2 + Ar]$ r = 0.7 and (**b**) $[N_2]$ r = 1.0.

On the other hand, the patterns of the samples deposited in the $N_2/Ar$ gas mixture (Figure 3b) showed the appearance of (100), (110), (111), (200), and (220) diffraction peaks. In this case, and at low gas pressures, a weak (100) peak emerged over an amorphous hump caused by the glass substrate. At the pressure of 3.5 Pa, the film showed a (111) preferred orientation, while in the diffraction pattern of the sample deposited at 5.0 Pa, the (100) plane appeared as the most substantial peak. Other authors previously observed this change obtained in orientation. It could be attributed to the higher density of the N atoms that reached the substrate and reacted with the Cu atoms, leading to high-density Cu-N bonds for the preferential growth along the (100) direction. This can indicate that the

nitridation was more effective when the gas pressure increased and, hence, the growth was favoured along the N-rich planes of $Cu_3N$ [38].

On the other hand, the constant lattice a was determined by calculating the interplanar spacing using Bragg's law [39], expressed as the following Equation (1):

$$d(hkl) = \frac{a}{\sqrt{(h^2 + k^2 + l^2)}} \tag{1}$$

where *d* is the interplanar spacing, and *h*, *k*, and *l* are the Miller indices. Furthermore, the grain size ($\tau$) was determined using the Debye–Scherrer Equation (2) [40] as follows:

$$\tau = \frac{k\lambda}{\beta \cdot \cos\theta_B} \tag{2}$$

where *k* is a constant (0.9), $\lambda$ is the X-ray wavelength (0.154 nm), $\theta$ is the diffraction angle, and $\beta$ is the full width at half maximum (FWHM) of the predominant peak.

Tables 2 and 3 summarise the FWHM of the main diffraction peak, the lattice constant, the predominant plane, the $2\theta$ value, and the grain sizes derived from the XRD patterns of all the samples, depending on the $N_2$ flow ratio.

**Table 2.** XRD data extracted from the XRD spectra of the $Cu_3N$ films fabricated on glass via RF magnetron sputtering in a mixed $N_2/Ar$ atmosphere.

| $N_2$ Flow Ratio: 0.7 | | | | |
|---|---|---|---|---|
| Working pressure (Pa) | 1.0 * | 2.0 * | 3.5 | 5.0 |
| $2\theta$ (°) | 23.03 | 22.97 | 40.73 | 23.23 |
| Predominant direction * | - | - | (111) | (100) |
| Lattice parameter a (nm) | 0.3858 | 0.3872 | 0.3813 | 0.3814 |
| FWHM (°) | 0.24 | 0.24 | 0.53 | 0.31 |
| Grain size (nm) | 34 | 34 | 16 | 27 |

* Poor crystalline quality. Preferential orientation is not easy to identify.

**Table 3.** XRD data extracted from the XRD spectra of the $Cu_3N$ films fabricated on glass via RF magnetron sputtering in a pure $N_2$ atmosphere.

| $N_2$ Flow Ratio: 1.0 | | | | |
|---|---|---|---|---|
| Working pressure (Pa) | 1.0 | 2.0 | 3.5 | 5.0 |
| $2\theta$ (°) | 22.77 | 23.17 | 23.29 | 23.28 |
| Predominant direction | (100) | (100) | (100) | (100) |
| Lattice parameter *a* (nm) | 0.3903 | 0.3840 | 0.3810 | 0.3813 |
| FWHM (°) | 0.20 | 0.15 | 0.21 | 0.21 |
| Grain size (nm) | 40 | 55 | 39 | 40 |

It can be observed that the FWHM values for the samples deposited in the $N_2/Ar$ gas mixed atmosphere were superior to those for the samples deposited in the pure $N_2$ atmosphere, indicating an improved quality for these last films. The same trend was obtained for the grain size, reaching values as high as 55 nm when the samples were fabricated in a pure $N_2$ atmosphere. Concerning the lattice parameter, at the low pressures of 1.0 and 2.0 Pa, greater values than the theoretical one (0.38170 nm) were achieved, regardless of the $N_2$ flow ratio used. This fact could indicate a move away from the stoichiometry condition for such samples [41].

Figure 4 pictures the plan-view FESEM and AFM $1 \times 1$ $\mu m^2$ 2D micrographs of the $Cu_3N$ thin films deposited in $N_2/Ar$ gas mixed atmosphere (Figure 4a) and pure $N_2$ environment (Figure 4b). FESEM analysis revealed the presence of smooth and uniform surfaces, composed mainly of columnar grains, which are characteristic of the sputtering method [32,42,43]. These findings align with the results obtained from the AFM analysis [34,44,45], as shown in Figure 4.

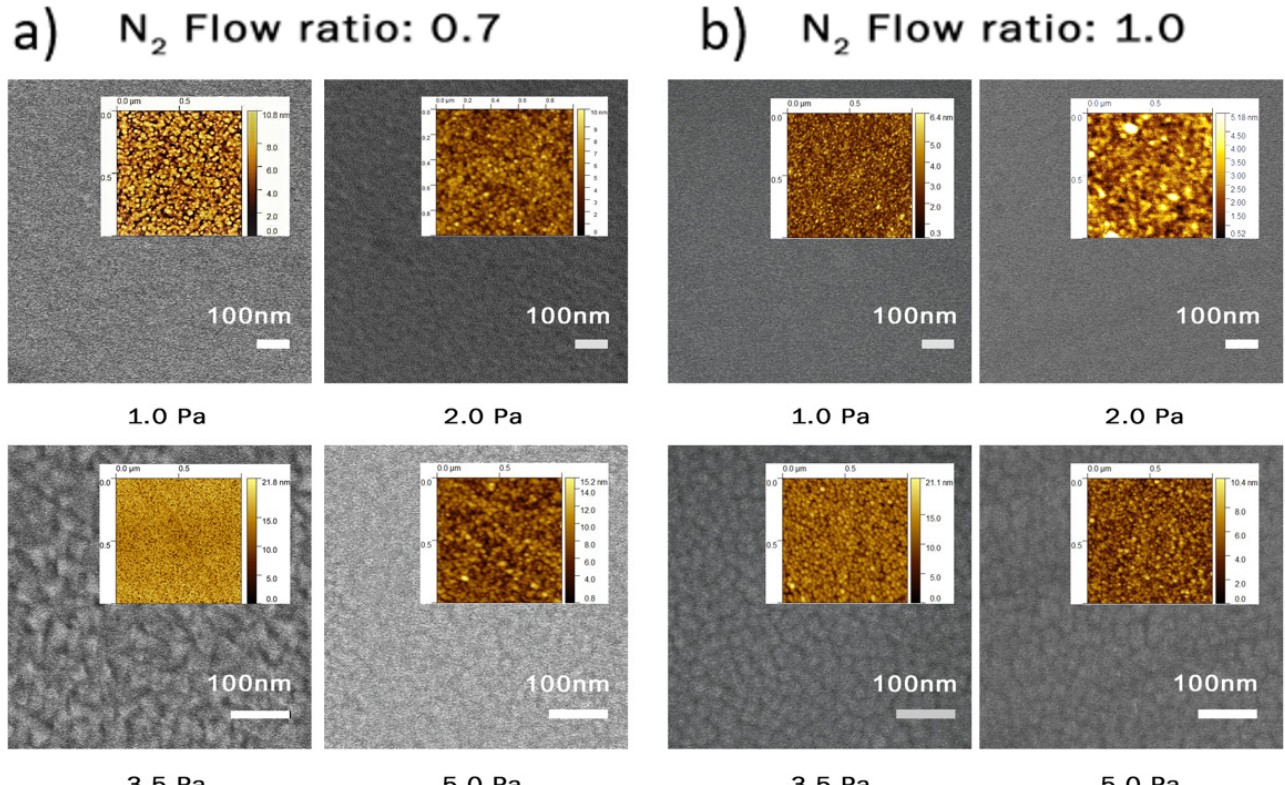

**Figure 4.** Top view FESEM images: Magn. 90,000×; 100 nm, and AFM $1 \times 1\ \mu m^2$ 2D micrographs. (**a**) $[N_2]/[N_2 + Ar]\ r = 0.7$ and (**b**) $[N_2]\ r = 1.0$.

The size of the grains was influenced by both the environment and the total pressure applied during the deposition process. It was observed that lower working pressures resulted in larger grain sizes. This phenomenon can be attributed to the formation of $Cu_3N$ crystallites and the adhesion of tiny copper crystals, possibly caused by a decrease in $N_2$ density. These observations are supported by grain size calculations performed using the commercial software Gwyddion. This result could be reinforced with the grain size values calculated with the Debye–Scherrer equation in XRD.

Tables 4 and 5 summarise the grain size and surface roughness RMS of the $Cu_3N$ films, calculated from the 2D AFM micrographs using the Gwyddion software, depending on the $N_2$ flow ratio.

**Table 4.** Surface roughness RMS and grain size calculated from AFM $1 \times 1\ \mu m^2$ images of the $Cu_3N$ films fabricated on glass via RF magnetron sputtering in a mixed $N_2/Ar$ atmosphere.

| $N_2$ Flow Ratio: 0.7 | | | | |
|---|---|---|---|---|
| Working pressure (Pa) | 1.0 | 2.0 | 3.5 | 5.0 |
| RMS (nm) | $1.39 \pm 0.15$ | $1.26 \pm 0.14$ | $2.77 \pm 0.23$ | $1.77 \pm 0.21$ |
| Grain size (nm) | $40 \pm 2$ | $35 \pm 3$ | $30 \pm 3$ | $34 \pm 3$ |

**Table 5.** Surface roughness RMS and grain size calculated from AFM $1 \times 1\ \mu m^2$ images of the $Cu_3N$ films fabricated on glass via RF magnetron sputtering in a pure $N_2$ atmosphere.

| $N_2$ Flow Ratio: 1.0 | | | | |
|---|---|---|---|---|
| Working pressure (Pa) | 1.0 | 2.0 | 3.5 | 5.0 |
| RMS (nm) | $0.90 \pm 0.15$ | $1.15 \pm 0.50$ | $2.21 \pm 0.27$ | $1.34 \pm 0.12$ |
| Grain size (nm) | $19 \pm 2$ | $32 \pm 2$ | $33 \pm 2$ | $35 \pm 2$ |

Based on these findings, it can be concluded that regardless of the total working pressure and $N_2$ flow ratio used, the surfaces were very flat, presenting RMS values that did not exceed 3 nm. In this sense, it can be noted that even though the RMS was very low, there was a difference between the films deposited using pure $N_2$ gas and those deposited in the $N_2/Ar$ gas mixture, obtaining slightly smoother surfaces in the first case. On the other hand, the grain size varied depending on the process parameters. Larger sizes as working pressure increased. The larger grain sizes obtained for the films deposited at 1.0 Pa and 2.0 Pa in a mixture of N/Ar gases may be due to the formation of small agglomerates due to the amorphous character seen in XRD. It should be pointed out that the grain size values estimated from XRD patterns were slightly higher than the obtained from AFM measurements. This can be explained because the first ones were an average value of a larger area analysed, while the second ones were calculated in a small area at a specific point.

The chemical composition of the samples was determined qualitatively using EDS data (Table 6). The analysis revealed a Cu/N ratio below three, indicating the non-stoichiometry of the deposited material. Interestingly, an increased Cu/N ratio was observed at higher working pressures. This phenomenon can be attributed to the increased energy of the nitrogen atoms at higher working pressures, enhancing the formation of bonds with the copper atoms. Moreover, all $Cu_3N$ films exhibited the presence of trace amounts of oxygen. This observation may be associated with exposure to ambient oxygen, as observed in the Raman analysis but not detected in XRD patterns.

**Table 6.** The EDS analysis provided insights into the relative surface composition of the examined films.

| Working Pressures | 1.0 Pa | 2.0 Pa | 3.5 Pa | 5.0 Pa |
|---|---|---|---|---|
| Cu/N ratio $[N_2]/[N_2 + Ar]$ $r = 0.7$ | 1.79 | 2.08 | 2.10 | 2.17 |
| Cu/N ratio $[N_2]$ $r = 1.0$ | 1.87 | 1.88 | 2.07 | 2.13 |

Figure 5 displays the Raman spectra of the $Cu_3N$ films deposited at different working pressures and $N_2$ flow ratios of 0.7 (Figure 5a) and 1.0 (Figure 5b). $Cu_3N$ has a crystal structure belonging to the Pm-3m space group where the unit cell contains one formula unit. As a result, no first-order Raman signal is expected for a perfect cubic $Cu_3N$. Although theoretical calculations suggest the absence of active Raman modes, the possibility of modes arising due to the breakdown of the selection rule cannot be ruled out. This is due to $Cu_xN_{(1-x)}$'s highly non-stoichiometric nature and the breakdown in crystal symmetry caused by defects in the structure. The prominent Raman peak around 619–627 cm$^{-1}$ in Figure 5 corresponds to the stretching of the Cu-N bond, characteristic of $Cu_3N$ [41,46,47]. A slight shift of that Raman peak was observed as the working pressure varied, while the Raman shift value tends to move to lower ones when using the $N_2/Ar$ gas mixture, compared to using the pure $N_2$ gas in the deposition process.

Furthermore, Raman shifts of $CuO_2$ and CuO also appeared in the spectra at 94 cm$^{-1}$, 150 cm$^{-1}$, and 295 cm$^{-1}$, respectively. The samples prepared in the $N_2/Ar$ gas mixture showed a higher presence of different copper oxides that may have formed on the film surface upon contact with atmospheric air and/or within the crystal structure. In order to analyse the Raman signal derived from the presence of these types of oxides, these measurements were complemented with the XRD data. As a result, it was confirmed that the characteristic $2\theta$ peaks of $CuO_2$ and CuO at 36.5° and 35.5°, respectively [48], did not appear in any diffraction patterns, reinforcing our prediction that the oxidation process would be happening due to environmental causes. However, the role of oxygen impurities cannot be ignored, as oxygen always remains an unintentional impurity in nitride-based materials [49]. Therefore, a more detailed theoretical analysis is required to interpret and assign the active Raman peak appropriately. Tables 7 and 8 show the prepared samples' Raman shift values and FWHM.

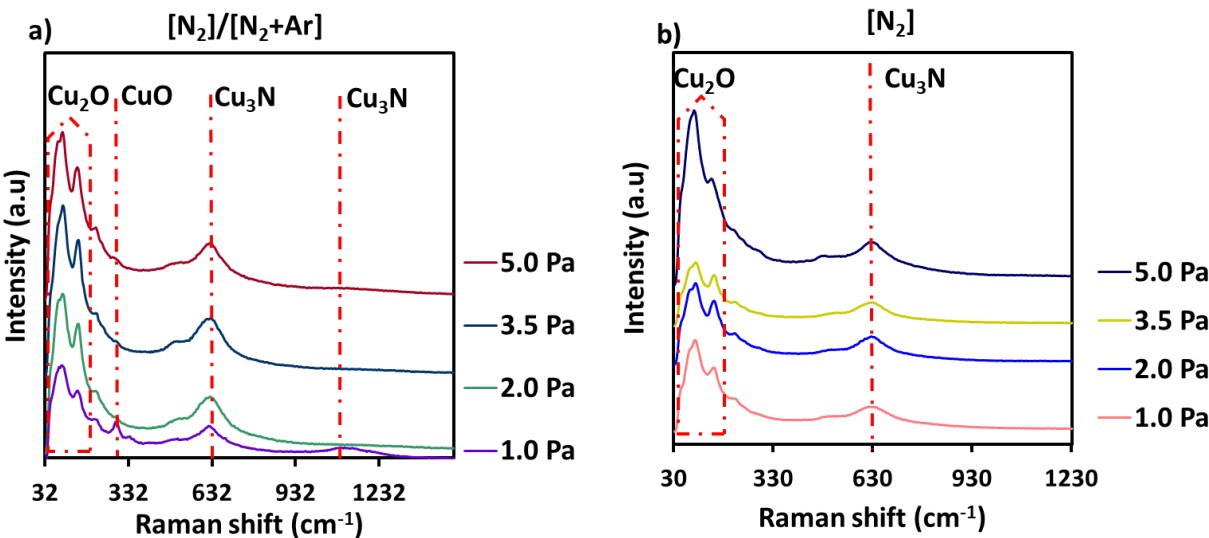

**Figure 5.** Raman spectra for $Cu_3N$ films deposited at different pressures on glass. (**a**) $[N_2]/[N_2 + Ar]$ *r* = 0.7 and (**b**) $[N_2]$ *r* = 1.0.

**Table 7.** Raman peak position and full width at half-maximum (FWHM) of the $Cu_3N$ films at different pressures on glass in a mixed $N_2/Ar$ atmosphere.

| $N_2$ Flow Ratio: 0.7 | | | | |
|---|---|---|---|---|
| Working pressure (Pa) | 1.0 | 2.0 | 3.5 | 5.0 |
| FWHM $(cm^{-1})$ | $78.9 \pm 2.8$ | $76.8 \pm 1.8$ | $82.9 \pm 3.4$ | $68.1 \pm 2.7$ |
| Peak position $(cm^{-1})$ | $618.0 \pm 1.2$ | $621.0 \pm 0.4$ | $619.0 \pm 0.6$ | $621.0 \pm 0.6$ |

**Table 8.** Raman peak position and full width at half-maximum (FWHM) of the $Cu_3N$ films at different pressures on glass in a pure $N_2$ atmosphere.

| $N_2$ Flow Ratio: 1.0 | | | | |
|---|---|---|---|---|
| Working pressure (Pa) | 1.0 | 2.0 | 3.5 | 5.0 |
| FWHM $(cm^{-1})$ | $82.9 \pm 1.7$ | $74.9 \pm 2.0$ | $68.0 \pm 3.6$ | $65.2 \pm 1.3$ |
| Peak position $(cm^{-1})$ | $623.0 \pm 0.3$ | $625.0 \pm 0.4$ | $626.0 \pm 0.5$ | $627.0 \pm 0.4$ |

The position of the main peak and the FWHM were calculated by simulating via the OriginLab program (OriginPro 8, OriginLab Corporation, Northampton, MA, USA). As observed, the FWHM value decreased as the working pressure increased, suggesting that the lower the FWHM value, the lower the nitrogen concentration in the sample. There is an exception for the sample prepared in the gas mixture at 3.5 Pa, where the FWHM did not exhibit that tendency, attributed to its structural change to the (111) preferential plane revealed by its XRD pattern. However, this sample does not follow that trend. In summary, these results obtained from XRD and Raman show that the films with superior quality that could serve as solar absorbers are those prepared in the pure $N_2$ atmosphere and at pressures of 3.5 Pa and 5.0 Pa. Furthermore, these films exhibited a less variety of grain orientations, with a predominant (100) plane and the Raman shift values were closer to the formation of the theoretical bonding structure.

Optical properties were determined from transmittance and reflectance spectra obtained using UV-Vis-NIR spectroscopy and shown in Figure 6. The transmittance spectra showed high transmittance in the NIR region (>700 nm), which gradually decreased in the VIS range (450–700 nm), reaching very low values in the UV range (300–400 nm).

For most samples, a minimum transmittance was observed in the transparent region (>700 nm), corresponding to a maximum reflectance. This is an apparent effect due to interference from multiple reflections inside the films because of the samples' homogeneity

and flatness, and, thanks to that. The light maintains its coherence in the internal reflections. It should be noted that the thin thickness of the films (<200 nm) limits the number of interferences observed in the spectra (in fact, the maximum transmittance beyond 2500 nm, which should be about 92% according to the refractive index of the glass, is not observed in any case). However, the detection of the minimum of $T_{opt}$ (or a maximum of $R_{opt}$) is sufficient to determine, by the fit of $T_{opt}$ (or $R_{opt}$) in the transparent region, the thickness and refractive index ($n_\infty$) of the films.

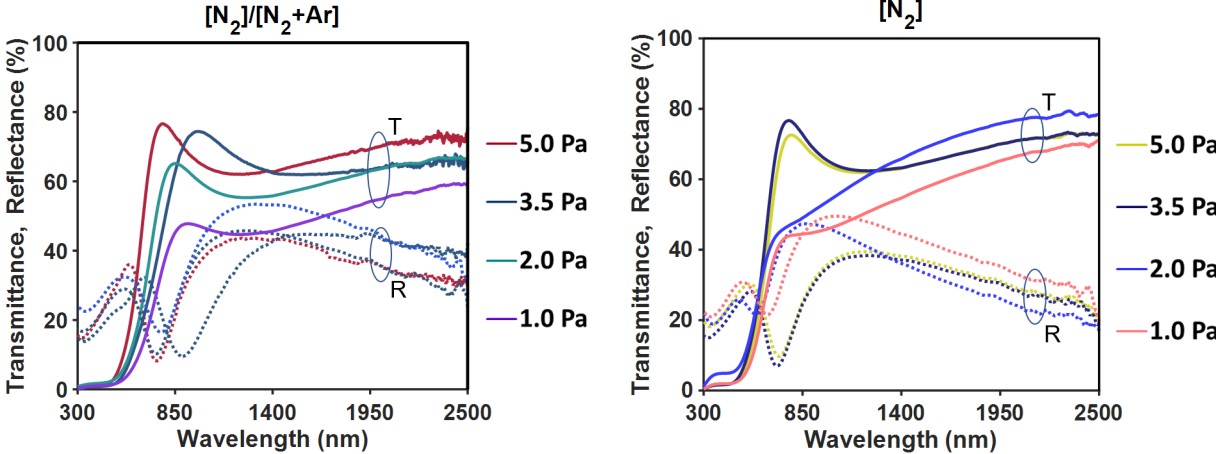

**Figure 6.** Transmittance and reflectance spectra of $Cu_3N$ films deposited at different pressures and atmospheres.

Table 9 summarises the optical parameters obtained. It can be observed that the thickness values obtained from the adjustments of the optical spectra were slightly superior to the measured with profilometry (data in Table 1). This disparity in thickness could be attributed to how this value is determined because profilometry provides direct thickness measurements, whereas PDS relies on indirect one.

**Table 9.** Obtained values of the optical properties of $Cu_3N$ films deposited on glass analysed via UV-VIS-NIR optical spectroscopy and PDS at different $N_2$ flow ratios.

| | $N_2$ Flow Ratio: 0.7 | | | |
|---|---|---|---|---|
| Working Pressure (Pa) | 1.0 | 2.0 | 3.5 | 5.0 |
| **Optical fit parameters** | | | | |
| Film thickness (nm) | 110 | 123 | 173 | 125 |
| Refractive index $n_\infty$ | 3.05 | 2.65 | 2.44 | 2.45 |
| **Band gap fit parameters** | | | | |
| Transition energy $E_2$ (eV) | 2.18 | 2.27 | 2.38 | 2.46 |
| Direct band gap $E_g^d$ (eV) | 1.91 | 2.02 | 2.09 | 2.22 |
| Indirect band gap $E_g^i$ (eV) | 1.10 | 1.28 | 1.22 | 1.53 |
| **Urbach fit parameters** | | | | |
| Transition energy $E_1$ (eV) | 1.55 | 1.69 | 1.64 | 1.90 |
| Urbach energy $E_U$ (meV) | 261 | 233 | 243 | 205 |
| Absorption coefficient at $E_1$ ($cm^{-1}$) | $6.3 \times 10^4$ | $5.1 \times 10^4$ | $4.0 \times 10^4$ | $5.2 \times 10^4$ |
| | $N_2$ flow ratio: 1.0 | | | |
| Working Pressure (Pa) | 1.0 | 2.0 | 3.5 | 5.0 |
| **Optical fit parameters** | | | | |
| Film thickness (nm) | 109 | 65 | 126 | 123 |
| Refractive index $n_\infty$ | 2.85 | 2.79 | 2.43 | 2.43 |
| **Band gap fit parameters** | | | | |
| Transition energy $E_2$ (eV) | 2.30 | 2.25 | 2.42 | 2.37 |
| Direct band gap $E_g^d$ (eV) | 2.05 | 2.06 | 2.21 | 2.15 |
| Indirect band gap $E_g^i$ (eV) | 1.30 | 1.48 | 1.55 | 1.49 |
| **Urbach fit parameters** | | | | |
| Transition energy $E_1$ (eV) | 1.74 | 1.82 | 1.96 | 1.88 |
| Urbach energy $E_U$ (meV) | 253 | 186 | 229 | 217 |
| Absorption coefficient at $E_1$ ($cm^{-1}$) | $8.0 \times 10^4$ | $6.5 \times 10^4$ | $6.0 \times 10^4$ | $5.3 \times 10^4$ |

The refractive index values were in the range of 2.4–3.0, which is consistent with the values typically reported for $Cu_3N$ [50]. It can be noticed that higher refractive index values were obtained for the samples deposited at lower total pressures, regardless of the atmosphere used in the film deposition. This phenomenon can be attributed to the higher Cu content in the samples deposited at lower working pressures. This assumption can also be supported by examining the XRD diffractograms and EDS analysis.

The absorption coefficient ($\alpha$) in the strong absorption region can be calculated with reasonable accuracy from the absorbance $A_{opt}$, reflectance $R_{opt}$ and thickness $d$ of the film as follows:

$$\alpha \approx -\frac{1}{d} \ln\left(1 - \frac{A_{opt}}{1 - R_{opt}}\right) \tag{3}$$

In the weak absorption region, Equation (3) is unsuitable for estimating $\alpha$ because of multiple reflections. However, in this case, assuming that the refractive index is practically constant ($n \approx n_\infty$), it is possible to derive $\alpha$ a wide range of the spectrum (from about 3.5 eV to 0.5 eV).

Figure 7a shows the $A_{opt}$ and $A_{pds}$ spectra for the $Cu_3N$ samples deposited at 5.0 Pa and $N_2/Ar$ gas mixture. As can be seen, the determination of $A_{opt}$ at wavelengths longer than 800 nm is unreliable (the error in the measurement of $T_{opt}$ and $R_{opt}$ is of the order of $A_{opt}$). However, the PDS measurement in this region allows us to obtain the absorbance, $A_{pds}$. Figure 7b shows the absorption coefficient spectrum obtained by combining the calculation according to Equation (3) for the intense absorption region ($\alpha > 1/d$) and the TMM model fit for the weak absorption region.

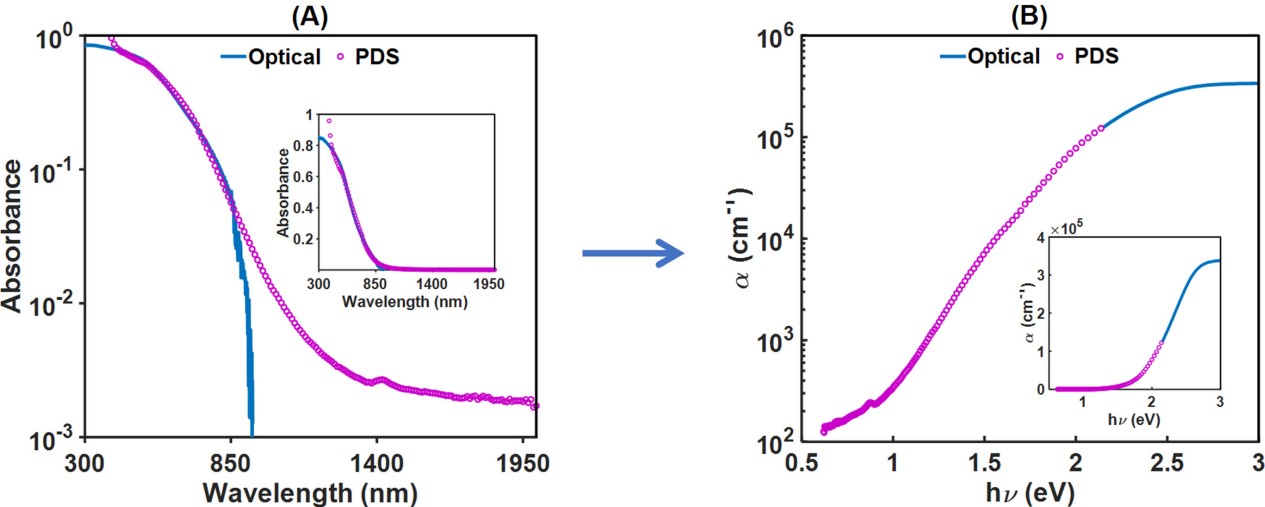

**Figure 7.** Absorbance obtained from optical measurements and PDS for the $Cu_3N$ film deposited at 5.0 Pa pressure in pure $N_2$ atmosphere. In (**A**), the absorbance spectra are shown, and in (**B**), the absorption coefficient is calculated from the fit based on the transfer matrix method. The insets show the same graphs on a linear scale.

Once $\alpha$ was obtained, the indirect and direct optical band gaps of $Cu_3N$ were obtained using the Tauc plot:

$$(\alpha h\nu)^{1/m} = B\left(h\nu - E_g\right) \tag{4}$$

where $h\nu$ is the photon energy, $E_g$ is the band gap energy, $B$ is a constant, and $m$ is a factor, which depends on the nature of the electron transition (2 for indirectly allowed transitions and 1/2 for direct allowed transitions). Figure 8 shows the Tauc plots with $m = 2$ and $m = 1/2$ of the different $Cu_3N$ samples deposited at different pressures in the two environments. The decrease in total pressure implies a decrease in both band gaps. This effect was more significant in the samples deposited on the $N_2/Ar$ mixture.

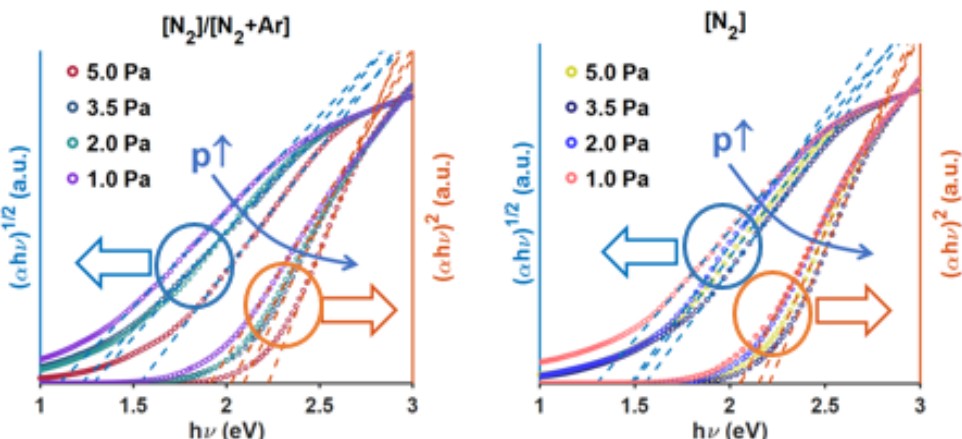

**Figure 8.** A plot of $(\alpha h\nu)^{1/2}$ and $(\alpha h\nu)^2$ vs. photon energy (eV) for the Cu$_3$N films at different gas pressures and atmospheres. For ease of comparison, normalised values of the absorption coefficient are considered.

Trying to further analysis, different energy ranges were distinguished according to the behaviour of $\alpha$. Thus, the electronic transitions associated with the direct band gap took place for photon energies in a narrow range from 2.1 to 2.4 eV, as a lower limit, to approximately 2.5 eV, as an upper limit. On the other hand, the electronic transitions associated with the indirect gap cover a larger range extending about 0.5 eV toward lower photon energies. In addition, at lower energies, exponential Urbach absorption ($\alpha_U$) can be observed, which is related to electronic transitions involving the band tails.

To accurately determine the direct and indirect band gaps, we performed a fine least-squares fit of the spectral dependence $\alpha$ according to the following model:

$$\text{At } h < E_1 : \quad \alpha_U(h\nu) = \alpha_0 \exp(h\nu / E_U) \tag{5}$$

where $E_1$ is the transition energy, $E_U$ is the Urbach energy (the slope of the exponential tail), and $\alpha_0$ is the absorption prefactor.

$$\text{At } h > E_2 : \quad \alpha_d(h\nu) = B_d \sqrt{h\nu - E_g^d} \quad \text{(direct Tauc model)} \tag{6}$$

$E_2$ is the transition energy, $B_d$ is a constant, and $E_g^d$ is the direct band gap energy.

$$\text{At } E_1 < h < E_2 \quad \alpha_i(h\nu) = B_i \left( h\nu - E_g^i \right)^2 \quad \text{(indirect Tauc model)} \tag{7}$$

$B_i$ is a constant, and $E_g^i$ is the indirect band gap energy. On the other hand, it can be observed that $\alpha$ must vary continuously and smoothly through the different regions. This implies imposing continuity conditions for $\alpha$ and its derivative at the two transition energies:

$$\begin{aligned} \alpha_i(E_1) = \alpha_U(E_1) \quad & \frac{d\alpha_i}{dE}(E_1) = \frac{d\alpha_U}{dE}\alpha_U(E_1) \\ \alpha_i(E_2) = \alpha_d(E_2) \quad & \frac{d\alpha_i}{dE}(E_2) = \frac{d\alpha_d}{dE}\alpha_U(E_2) \end{aligned} \tag{8}$$

which reduces the eight parameters of the model described by Equations (5)–(7) to only four independent ones. The graphs in Figure 9 depict the fit of the experimental data with the model described by Equations (5)–(8). A correlation between these energies and total pressure was observed, the effect more measurable for the samples deposited in the mixed N$_2$/Ar atmosphere. The inset in Figure 8 pictures the fit in the sub-gap region at the edge of the indirect band gap at $E_1$, plotted in a logarithmic scale. This is the region where the absorption coefficient is described by the Urbach exponential, related to the states in the band tails. The same as was deduced from the Tauc plots in Figure 7, the samples with the narrowest band gap (direct and indirect) were obtained at the lowest total pressures.

Therefore, by modifying the deposition parameters, such as total working pressure and $N_2$ flow ratio, the optical properties of $Cu_3N$ thin films can be tailored to achieve desirable properties, making them promising for applications in optoelectronics and photonics.

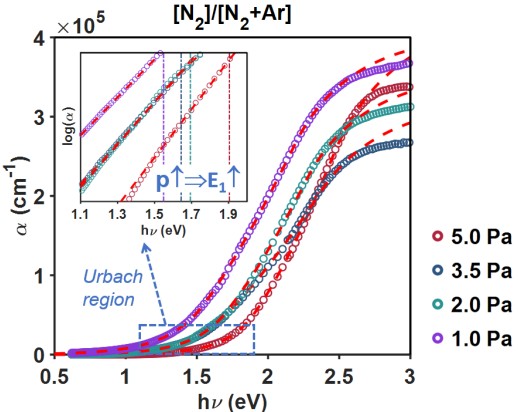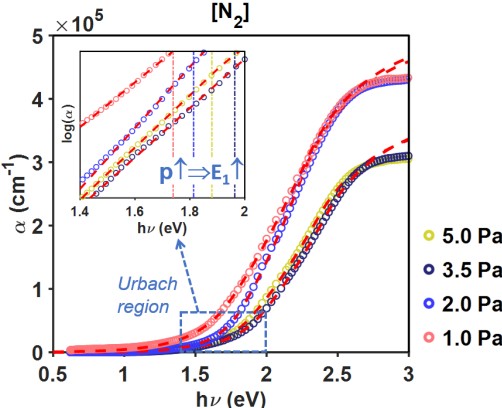

**Figure 9.** Absorption coefficient spectra of $Cu_3N$ layers at different pressures and atmospheres. The red dashed lines show the fit according to the model equations described (5)–(8). In the logarithmic scale, the insets show the exponential tail of the Urbach region; note the effect of pressure on the transition energy $E_1$.

Finally, the shaded regions in Figure 10 represent the range of photon energies ($E_1 < h\nu < E_2$) associated with the electronic transition by the indirect gap. In general, there is an increasing tendency for the band gap with the total pressure.

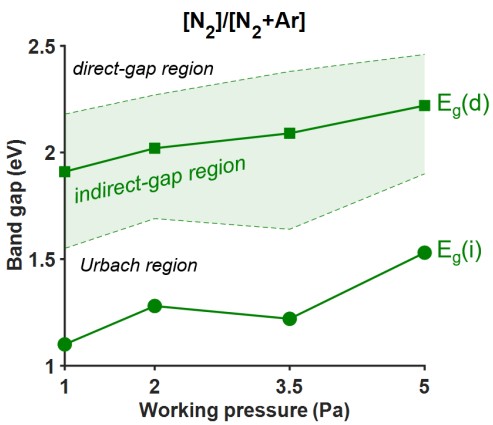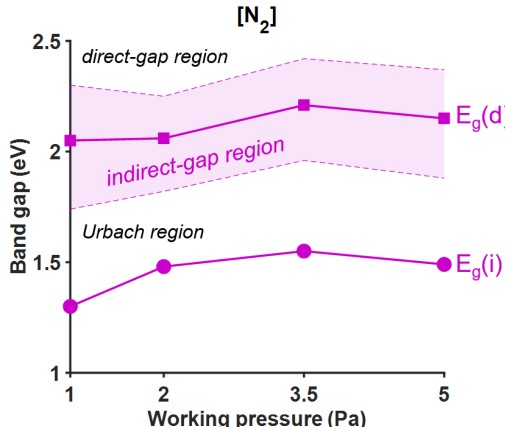

**Figure 10.** Evolution with a working pressure of the energies of the direct gap Eg(d) and the indirect gap Eg(i) for the two deposition atmospheres. The shaded region indicates the range of photon energies ($E_1 < h\nu < E_2$) associated with electronic transitions according to the indirect gap: for higher energies ($h\nu > E_2$), the transitions are associated with the direct gap and for lower energies ($h\nu < E_1$) with the Urbach tail.

Regarding the Urbach energy ($E_U$), increasing values were obtained as the total pressure decreased. These values were superior to those of other semiconductor materials [20]. It is known that high $E_U$ values indicate a higher internal defect density, while lower Eu ones suggest a lower internal defect density. Hence, the Urbach band tail is related to impurity adsorption [51,52]. According to the literature, $Cu_3N$ shows $E_U$ values ranging from 105 to 238 meV [20,51] depending on the substrate temperature. Therefore, it can be concluded that the samples deposited at higher total pressures feature reasonable band gap energies and $E_U$ values to be used in photovoltaic applications.

## 4. Conclusions

In this study, the $Cu_3N$ films were successfully deposited on glass substrates via RF magnetron sputtering using a power of 50 W and different $N_2$ flow ratios to determine their effect on the film properties. The results provide valuable insights into these films' optical and structural properties. The analysis of the film structure revealed polycrystallinity, with a preferred growth orientation along the (100) plane when the $N_2$ flow ratio was $r = 1.0$. However, an amorphous matrix was observed for an $N_2$ flow ratio of 0.7 and low total working pressures up to 2.0 Pa; at working pressures above 2.0 Pa, a tendency for growth along the (111) plane was obtained. These findings and the lattice parameter values extracted from the XRD patterns suggest that the films deposited in the $N_2/Ar$ gas mixture tend to have a higher concentration of copper. Raman characterisation confirmed the formation of Cu-N bonds, as evidenced by the characteristic peak observed in these spectra. The presence of oxygen in the Raman spectra and in the EDS analysis was attributed to environmental factors due to no Cu-O bond-related structures detected in the XRD patterns. The Cu/N ratio demonstrated an increase in the total working pressure. FESEM and AFM analysis showed a film morphology consisting of columnar grains with a very smooth and homogeneous surface. Through a combined analysis of the optical properties using conventional UV-VIS-NIR and PDS spectroscopies the absorption coefficient over a wide range of photon energies (from 0.5 eV to 3.5 eV) was determined. A model with two band gaps, indirect and direct, and the Urbach exponential tail in the sub-gap region described the complete absorption coefficient spectrum. Depending on the deposition conditions, the energy of the direct gap varied in the range of 1.1 to 1.5 eV and the direct gap in the range of 1.9–2.2 eV. Generally, as the working pressure decreased, the energies of the two gaps tended to decrease, with the effect being most evident in the layers deposited in the $N_2/Ar$ gas mixture environment. The samples deposited with the lowest working pressure (1 Pa) presented the highest value of the Urbach energy (>250 meV). The minimum value of the Urbach energy of 183 meV was found for the sample deposited at 2 Pa in a pure $N_2$ atmosphere. Finally, these films exhibited desirable structural, morphological, and optical properties, making them promising candidates as solar absorbers. The findings could contribute to developing and optimising $Cu_3N$-based materials for efficient solar energy conversion.

**Author Contributions:** Conceptualization, M.I.R.-T. and S.F.; methodology, M.I.R.-T. and S.F.; software, J.M.A.; validation, M.I.R.-T., M.R. and J.M.A.; formal analysis, M.R., J.M.A., J.M. and M.I.R.-T.; investigation, M.I.R.-T. and S.F.; resources, J.B. and S.F.; data curation, M.I.R.-T., M.R. and J.M.A.; writing—original draft preparation, M.I.R.-T., M.R. and J.M.A.; writing—review and editing, M.I.R.-T., M.R., J.M.A., J.B. and S.F.; visualization, M.I.R.-T., J.M.A. and M.R.; supervision, S.F.; project administration, J.B. and S.F.; funding acquisition, J.B. and S.F. All authors have read and agreed to the published version of the manuscript.

**Funding:** This research was funded by MCIN/AEI/10.13039/501100011033, grant number PID2019-109215RB-C42 and PID2019-109215RB-C43. M.I.R.-T. also acknowledges partial funding from ME-DIDA C17.I2G: CIEMAT. Nuevas tecnologías renovables híbridas, Ministerio de Ciencia e Innovación, Componente 17 "Reforma Institucional y Fortalecimiento de las Capacidades del Sistema Nacional de Ciencia e Innovación". Medidas del plan de inversiones y reformas para la recuperación económica funded by the European Union—NextGenerationEU.

**Institutional Review Board Statement:** Not applicable.

**Informed Consent Statement:** Not applicable.

**Data Availability Statement:** Not applicable.

**Acknowledgments:** The authors would like to thank A. Soubrié from Centro de Microscopía Electrónica "Luis Bru" for her help and advice in AFM measurements.

**Conflicts of Interest:** The authors declare no conflict of interest.

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
