# Peer review of "Copper Nitride: A Versatile Semiconductor with Great Potential for Next-Generation Photovoltaics"

_coatings, doi:10.3390/coatings13061094_

Round 1

Reviewer 1 Report

This manuscript with title of Copper nitride: a versatile semiconductor with great potential for next-generation photovoltaics looks systematically organized well. Notably, the characterization of the material is solid. However I recommend that minor revision is strongly needed for publishing this journal due to the major issue like the below.

1.  To clarify this study, the manuscript should be included the performance of the solar cell by implementing a simple device.

2. The images for SEM and AFM should be enhanced with high quality of over 300 dpi.

Hope to see the revised manuscript.

Author Response

The authors would like to thank you, for your valuable comments and suggestions.

Reviewer 2 Report

Copper nitride: a versatile semiconductor with great potential for next-generation photovoltaics" deals with the synthesis of copper nitride via sputtering. Following are my suggestions for the authors:

1. Line no. 308 .. it should be XRD and not RDX. There are many such typo errors like one more in line no. 511.

2. Fig 8.... authors stated that the material is direct and indirect both. How?

3. Table 1....With increase in sputtering pressure deposition rate generally goes down. But in table 1 there is abrupt increase in thickness for sample deposited in 3.5 Pa in N2 + Ar atmosphere.  This needs a better explanation supported with references. 

4.  Figure 3... The sample deposited in 2 Pa pressure shows a left shift. If it is a peak over the amorphous glass hump then why a peak shift. Explanation needed.

5.  In EDS pl. mention the individual at% of Cu and N. Also only EDS is not sufficient to affirm the composition. Some other techniques like XPS etc may be employed.

6. Line 319... authors stated that Cu3N has highly non-stoichiometric nature then please mention it like Cu3xN(1-x).

7.  FTIR spectroscopy can be done to measure the effectiveness of Cu3N as a photovoltaic material by measuring the emissivity.

There are several typo errors so the authors are requested to give it a proof read.

Author Response

The authors would like to thank you, for your valuable comments and suggestions

Round 2

Reviewer 2 Report

I am satisfied with the reply provided. The manuscript may be allowed to accept in its present form.